# Design and Development of a Device (Sifilotto^®^) for Tumour Tracking in Cervical Cancer Patients Undergoing Robotic Arm LINAC Stereotactic Body Radiation Therapy Boost: Background to the STARBACS Study

**DOI:** 10.3390/curroncol32060354

**Published:** 2025-06-16

**Authors:** Silvana Parisi, Giacomo Ferrantelli, Anna Santacaterina, Elvio Grazioso Russi, Federico Chillari, Claudio Napoli, Anna Brogna, Carmelo Siragusa, Miriam Sciacca, Antonio Pontoriero, Giuseppe Iatì, Stefano Pergolizzi

**Affiliations:** 1Department of Biomedical Dental and Morphological and Functional Imaging (BIOMORF), Residency School of Radiation Oncology, University of Messina, Via Consolare Valeria 1, 98124 Messina, Italy; silvana.parisi@unime.it (S.P.); anna.santacaterina@virgilio.it (A.S.); elviorussi@gmail.com (E.G.R.); cld.napoli@gmail.com (C.N.); miriamsciacca06@gmail.com (M.S.); antonio.pontoriero@unime.it (A.P.); giuseppe.iati@unime.it (G.I.); stpergolizzi@unime.it (S.P.); 2Department of Discipline Chirurgiche, Oncologiche e Stomatologiche (Di.Chir.On.S.), PhD School “Precision Medicine”, University of Palermo, Piazza Marina 61, 90133 Palermo, Italy; 3SDOU Radiation Oncology, University Hospital “Papardo”, Contrada Papardo, 98158 Messina, Italy; 4Radiotherapy Unit, “Villa Santa Teresa Diagnostica Per Immagini e Radioterapia”, Viale Ing. G. Bagnera 14, 90011 Bagheria, Italy; chillari.federico92@gmail.com; 5“D.A.I Servizi” Department, COU Medical Physics, University Hospital “AOU G. Martino”, Via Consolare Valeria 1, 98124 Messina, Italy; anna.brogna@polime.it (A.B.); carmelo.siragusa@polime.it (C.S.); 6Department of Clinical and Experimental Medicine (DIMED), University of Messina, Via Consolare Valeria 1, 98124 Messina, Italy

**Keywords:** cervical cancer, brachytherapy, endovaginal applicator, locally advanced, stereotactic, robotic LINAC

## Abstract

In the treatment of locally advanced cervical cancer, brachytherapy is often a crucial step after external beam radiation and concurrent chemo-immuno-radiation. However, when brachytherapy is not feasible due to patient or tumor-related challenges, delivering a precise external beam boost becomes necessary. Traditionally, this requires invasive implantation of fiducial markers to guide treatment when a robotic LINAC is used. To offer a non-invasive alternative, we developed a novel 3D-printed intravaginal device containing embedded gold fiducials for accurate tumor tracking during stereotactic radiotherapy with a robotic arm LINAC. This innovation eliminates the need for surgical procedures, improving patient comfort and workflow efficiency. This paper details the design and implementation process of the device, which led to a utility model patent.

## 1. Introduction

Squamous cervical carcinoma (SCC) is the most common histological type of cervical cancer. It is the third leading cause of cancer death in women, following breast and lung cancers. SCC is closely associated with HPV infection and the prognosis depends both on the stage and its biological features. If the disease is diagnosed at a locally advanced stage, the prognosis is poor, with a 5-year survival rate of 25–30% [1]. About 30% of patients with cervical cancer present a locally advanced, unresectable, disease (Locally Advanced Cervical Cancer, LACC) at diagnosis. Nowadays, a therapeutic approach, in this setting of patients, is concurrent chemoimmunoradiotherapy (cCIRT) followed by intracavitary brachytherapy (ICBT) with radical intent and immunotherapy maintenance. Further, in fact, an important study has confirmed superior overall survival clinical results obtained using immunotherapy in this clinical scenario (Keynote A18) [2].

Furthermore, when immunotherapy was not still available, concurrent chemoradiotherapy (cCRT) plus brachytherapy has been shown to improve local control (LC) [3]. This treatment modality has demonstrated a non-inferiority profile in terms of clinical outcomes when compared to neoadjuvant chemotherapy (NACT) followed by anterior pelvic exenteration, improving the results of toxicity and quality of life [4]. Regarding the neoadjuvant strategy, it is worth noting that another important study (international, multicentre, randomised phase III trial) investigated the role of NACT (which in this case may be more accurately referred to as “induction chemotherapy”) by administering neoadjuvant chemotherapy with six cycles of weekly carboplatin (AUC2) and paclitaxel (80 mg/m^2^) before cCRT, resulting in improved PFS and OS when compared to cCRT alone. Such neoadjuvant chemotherapy might make the subsequent brachytherapy easier [5]. The intracavitary brachytherapy approach involves the use of a three-way applicator consisting of a central tandem and two lateral and simmetrical ovoids, left and right, respectively. The tandem is positioned within the uterine cavity, while the ovoids are placed within the lateral vaginal fornices. Despite the fact that this technique represents the established therapeutic standard, it has some boundaries arising from its technical characteristics, including anatomical variants of the uterus, cervix and vagina, vaginal trophism influenced by the patient’s age, and the occurrence of vaginal stenosis resulting from prior treatments. Additionally, the persistence of endoluminal disease in the cervix can impede the passage of the tandem, thereby compromising the efficacy of the treatment itself.

Consequently, some studies report a boost to the cervix using external beam radiotherapy (EBRT), with doses and techniques that could substitute intracavitary treatment [6]. Favourable outcomes, particularly regaarding local toxicity and adherence to doses to organs at risk (OaR) were observed with the utilisation of stereotactic body radiotherapy (SBRT) [7], but there is no agreement about the best technical approach regarding LINAC type nor best procedures.

This study presents an endovaginal applicator designed and produced to optimise the therapeutic offer for patients with LACC that is unsuitable for intracavitary brachytherapy and who are selected to undergo stereotactic radiotherapy delivered with a robotic-arm LINAC.

This applicator has obtained a utility model patent.

## 2. Materials and Methods

### 2.1. Aims of Invention

A list of design requirements for the 3D-printed endovaginal applicator was developed through consultation with a group of radiation oncologists, physicists, and a mould room technician. The applicator was required to meet several criteria, including the capacity for a reproducible setup, biocompatibility, durability to withstand degradation during treatments, ergonomics, ease of cleaning (common germicide solutions) and maintenance, sustainability and cost-effectiveness. The proposed design was formulated based on these criteria, as well as a thorough review of the bioengineering literature data. The development process, which incorporated design, prototyping, consultation, and refinement, was undertaken to create a design that addressed the core-project group’s identified design requirements.

The 3D design process was executed utilising the Autodesk-Tinkercade software^®^ (https://www.tinkercad.com/dashboardc, accessed on 13 June 2023). The designs were then printed using a FLSUN V400-3D-printer^®^ (FLSUN-Zhengzhou Chaokuo Electronic Technology Co., Zhengzhou, China, https://flsun3d.com/pages/about-us, accessed on 13 June 2023).

The material selected for the primary component was polylactic acid (PLA), a biodegradable, biocompatible, and strong material that has been successfully employed in other radiotherapy applications and has shown resistance to radiation damage [8,9,10,11,12].

### 2.2. Design

To study and design the applicator, a primordial prototype was first created. The first prototype was a thin transparent plastic tube, 3 cm in diameter and 10 cm in length, filled with a mouldable wax-based compound in order to guarantee radiolucency. The fiducials were embedded within the wax according to a precise spatial arrangement before insertion into the transparent plastic cylinder. Several image acquisitions and fiducial positioning tests were carried out on this prototype to determine the best spatial arrangement.

We outlined some specific spatial characteristics for the applicator: Each fiducial should be positioned at least 12 mm from the centre of the target;A minimum fiducial spacing of 18 mm;An inclination of 45° with respect to the horizontal axis of the target centre;An inclination of at least 15° with respect to the long axis of the fiducial.

Figure 1 shows an example of the spatial positioning of the fiducials selected in the prototype study phase. The target centre is indicated with 1, and the fiducial positions are indicated by 2, 3, 4 and 5.

Based on these assumptions and tests, the digital drawings of the updated device were then developed.

Four drawings corresponding to 4 applicators were digitally created, considering the potential inter-patient anatomical variability of the vaginal canal and to improve the anatomical adaptability of the applicator itself. These were designed with different diameters of 35 mm (Figure 2), 30 mm (Figure A1), 25 mm (Figure A2) and 20 mm (Figure A3), respectively. Each applicator consisted of a cylinder with a round-shaped top (body), a neck and an applicator stopper, which also housed the urinary catheter. Figure 2 shows detailed components of the device: body, 9.5 cm (A); cylinder neck, 2.5 cm (B); and applicator stopper (C) with housing for the urinary catheter (D).

### 2.3. Fabrication

A biocompatible PLA filament was chosen for the 3D print. The chosen filament is 100% recycled, eco-sustainable, and composed of lactic acid produced by the fermentation of starch during the compostable wet milling of corn and rice. Technically, the filament has a diameter tolerance with a precision of 1.75 mm ± 0.02 mm, tensile strength ≥ 55 MPa and a hardness HRC of 105–110.

Fiducial housings were then created in each applicator using a pen soldering iron to a predetermined depth. There were 4 of these in each applicator at depths from the applicator surface of 12 mm for 20 mm and 25 mm applicators, and 17 mm for 30 mm and 35 mm applicators.

To select the position, the distances from the centre of the applicator were calculated using a trigonometric formula: the vertical axis was traced in the direction of the vertex-base of the applicator and then the perpendicular bisector was traced. The two perpendicular bisectors thus defined four 90° angles. By calculating the bisectors of the angles to obtain the respective axes of symmetry of the angles (α/2), two equal amplitudes (45°) were obtained.

On the axis of symmetry of the angle, with respect to the centre, the distances for the fiducial housings were measured: in the 35 mm and 30 mm cylinders, the housings were created at 20 mm and 15 mm from the centre, on the upper and lower faces of the cylinder placed in a horizontal position (4 in total); in the 25 mm and 20 mm cylinders, the fiducial housings were positioned at 16 mm and 12 mm from the centre, respectively, on the upper and lower faces of the cylinder placed in a horizontal position (4 in total). Each housing was created with an inclination of approximately 15° tangential to the surface of the applicator because of the oval shape of the gold fiducial itself.

Once the fiducials had been inserted using a mandrel guide, the housings were closed with the same PLA filament using a MYNT3D-pen^®^ (Mynt3d, Salt Lake City, UT, USA, https://www.mynt3d.com/pages/contact, accessed on 13 June 2023).

Figure 3 and Figure 4 show the finished 30 mm model and the four finished models, respectively.

## 3. Results

Radiological and clinical evaluation of the device have been performed following a step-by-step process. Validation of one step was a fundamental parameter for the next one. Firstly, we acquired radiological images for each device and verified the expected radiolucency due to predicted densitometric features. Also, we evaluated the good quality of the image acquisition of gold fiducials even if they were included inside the applicator. After that, we tested Sifilotto^®^ for clinical application in patients with LACC who underwent cCRT and who are not able to receive ICBT (see Section 1, lines 61–66) and need to receive a “completion” boost dose according to the STARBACS (see Section 6 “Patent”, “Institutional Review Board Statement”) with excellent results in terms of alignment, tracking and patients’ compliance.

### 3.1. Radiological Tests

Step 1. “CT test”: This involved the acquisition of CT simulation scans with the applicator positioned horizontally (as for endovaginal insertion) on a radiolucent solid water base and transmission of the images to the Treatment Planning System (TPS). During Step 1, the following were verified: correct insertion and positioning of the fiducials in the slots, verification of the rigid body and fiducial interval parameters. Figure 5 shows the CT images acquired and displayed on the TPS during the verification of the above parameters.Step 2. “x-ray imaging test”: This step includes the acquisition of verification x-ray images with the applicator positioned horizontally on a radiolucent solid water base. In Step 2, the x-ray images were acquired using two cameras located inside the bunker, on the right and left of the bed, at 190 cm from the floor and at an angle of 45° with respect to the floor. Thanks to the correspondence check with the DRR on the TPS, it was verified that the rigid body and fiducial interval parameters were correctly respected. In Step 2, it was also verified that both cameras clearly detected all four fiducials without any interference from the PLA material used in the construction. Figure 6 shows the x-ray image captured and displayed on the console monitor during the control of the parameters described above.

Both applicator tests were successful, and no further modifications were required.

### 3.2. In Vivo Tests

The endovaginal applicator was tested in a group of seven patients.

All patients had initially been diagnosed with locally advanced squamous cell carcinoma of the cervix. Prior to the commencement of the trial, each patient underwent pretreatment staging examinations, which included contrast-enhanced chest-abdomen CT, pelvic MRI, and 18-FDG-PET-CT.

All seven patients enrolled in the in vivo testing of the applicator had undergone cCRT with weekly cisplatin of 40 mg/mq and a total dose delivered to the Clinical Target Volume (CTV) of 45–50 Gy and simultaneous integrated boost (SIB) to positive pelvic lymph nodes of 55 Gy, if there were any. Only one patient had undergone large-field treatment also including lumbo-aortic lymph nodes, with a total dose delivered to the lumbo-aortic lymph nodes of 45 Gy according to an EMBRACE analysis [13]. Prescription doses regarding SBRT boost are beyond the scope of this paper and will be addressed in detail upon completion of the STARBACS protocol, although it is worth noting that the cumulative dose to achieve after EBRT + ICBT in the most important clinical trials is a minimum of a 2 Gy equivalent dose of 78–86 Gy [14] and coverage doses are not compromised on the target volumes without variations in doses on organs at risk.

The enrolled patients underwent pelvic MRI at the end of cCRT and commenced complementary therapy within 8 weeks.

Of the seven patients, two were deemed ineligible for intracavitary treatment during the clinical visit due to the presence of grade 2–3 vaginal stenosis and concomitant vaginal atrophy resulting from oestrogen deprivation that had persisted for over a decade. One patient was deemed ineligible for intracavitary treatment due to the inadequate response of cervical barrel disease that had extended to the upper third of the vagina, thereby impeding the passage of the tandem. Another patient was deemed ineligible due to the persistence of disease in the left parametrium and the infiltration of the posterior bladder wall. Additionally, three patients declined intracavitary brachytherapy.

Following a clinical evaluation of the vaginal diameter, the most suitable applicator was identified for each patient. The treatment of three patients was conducted using a 20 mm applicator, two patients were treated using a 25 mm applicator, and two patients were treated using a 30 mm applicator.

The patients underwent a CT simulation scan with the endovaginal applicator after insertion of a Foley bladder catheter and rectal toilet performed by rectal microenema plus daily insertion of hyaluronic acid. During the setup and treatment phases, the bladder was filled via catheter with 60 mL of saline solution, and Foley clamping was employed to ensure reproducibility during treatment. Following the completion of setup, the bladder was emptied, and the device was extracted. During the in vivo verification phase, a double alignment system was provided for the machine: “Spine Tracking” (for verification of anatomical positioning based on the column) and “Fiducials Tracking”.

During the in vivo trials conducted on each patient, the subsequent technical parameters were the focus of the evaluation: the six degrees of rotation, the length of translations involving their potential manual correction during patient setup, the possible rigid body errors, the fiducial interval and total treatment time were evaluated. Furthermore, “subjective” parameters related to the treatment with the applicator were evaluated through a test where the patient was asked to assign a score from 0 to 10 regarding pain, burning, discomfort in keeping the applicator in place during CT and during treatment, and psychological impact of the treatment. The subsequent analysis of the results obtained revealed that, reagarding the technical parameters extracted during alignment, the average time for image acquisition and positioning was reduced (5 min. vs. 15 min. in patients treated with transmucosal implantation of the fiducials).

In the present study, no cases of rigid body error or insufficiency of the fiducial interval were detected.

The most intriguing data acquired during in vivo tests pertains to the rototranslational parameters. As previously mentioned, during the daily (plus intrafraction) acquisition of X-ray imaging and comparison with the planning DRRs, six degrees of rototranslation of the bed and therefore of the patient were evaluated. Of these six rototranslations, only one was deemed to be potentially amenable to manual correction during alignment within the machine. This was the ‘vertex-feet’ inclination of the device (coronal plane), which, when necessary, could be rectified through minimal manual pitching movements of the vertex-feet of the applicator placed in position (Figure 7).

These data confirm that the invention successfully passes the reproducibility and optimisation tests of the setup in patients with locally advanced gynaecological tumours that are not eligible for intracavitary treatments.

Assessments are currently being conducted for late toxicities and clinical outcomes, for which a more protracted observation period is required.

Regarding the subjective parameters, two patients assigned a score of 7 to the burning symptom and ≤5 to pain; five patients assigned a score of ≤5 to both symptoms. Discomfort and psychological impact were both given a score of <5 by all patients (Table 1).

## 4. Discussion

### 4.1. Robotic Arm Linear Accelerators (R-LINACs)

Robotic arm linear accelerators (r-LINACs) are designed for image-guided stereotactic radiotherapy and radiosurgery. There is a undervalued variety of robotic systems all around the world, but Cyber Knife ^®^ is probably the most popular [15], whose role has already been investigated in some previous clinical studies for recurrent cervical cancer [16]. Furthermore, other previous experiences regarding the combination of SBRT as a completion boost dose after cCRT in locally advanced stages of cancers have been evaluated [17,18]. By exploiting the spatial non-coplanarity of the irradiation beams, a better dose conformation to the target is allowed, leading to better coverage while guaranteeing an extreme dose fall-off within 0.5 cm around the target volume respective to C-arm LINACs, thus sparing organs at risk (OaR), and used in very critical sites like CNS structures. Treatments with r-LINACs on pelvic organs necessitate the utilisation of intramucosal implanted gold fiducials for an accurate setup and positioning of patients throughout the entirety of the radiation treatment. These fiducials are typically implanted within the tumour or within the nearest surrounding tissues. The Treatment Planning System (TPS) is thought to be able to recognise the gold fiducials (due to the high atomic number, Z) as 3D spatial references basing on interspersed online radiographies generated by two 45°-oblique x-ray tubes (1 left, 1 right). This feature allows us to determine needed translations and rotations for the best patients’ position and highly accurate delivery to the target. The tracking algorithm calculates the deviation between online x-ray images and digital reconstructed radiographies (DRRs) by identifying and matching reference points in the images. This system is called “Synchrony Fiducial Tracking” mode and it is designed for the identification and tracking of implanted fiducials.

The tracking algorithm is then translated to numerical information and displayed on the deviations panel, allowing the clinician to apply the suggested movements on the treatment table. The deviations panel displays the following information:Axis: Indications of translations and rotations. These are represented on the screen with arrows pointing in a positive or negative direction on icons of the human body and corresponding to the patient’s orientation in the treatment plan.Calculated: Deviations (in millimetres or degrees) between X-ray imaging and DRR calculated by the tracking algorithm.Applied: Correction movements sent to the processing machine. These values can be verified by re-acquiring the image or can be directly displayed during treatment administration in the delivery phase.

The fiducials, which are inserted into the tissue, show a spatial migratory tendency from the insertion time until the tissue accommodation time is reached.

Despite the technique’s accuracy and its capacity to offer enhanced OaR savings, it is important to note that it is subject to limitations, including the following:The invasiveness of the fiducial insertion procedure;The risk of error in treatment reproducibility due to the migratory tendency of fiducials;The increase in the total treatment time due to aligning difficulties given by the intramucosal implant;The high costs due to the use of at least four fiducials per patient (cost of a single fiducial: EUR 55–150).

### 4.2. Costs Analysis

Cost reduction is one of the advantages that would result from the use of Sifilotto^®^ in daily clinical practice. In fact, with the intramucosal implant process, the fiducials must be inserted into each patient and once the therapeutic process is complete, they remain implanted for life. It is noteworthy that each fiducial (contained in a guide needle with a mandrel) has a variable cost between EUR 50 and EUR 150 and, to obtain a good evaluation of the rototranslations, at least four of them must be inserted into each patient.

The cost of each fiducial must be added to the total cost of the procedure, which includes hospitalisation of the patient in a protected hospital environment (hospital DRG), premedication for the interventional insertion session with the use of local anaesthetic, infusion of fluids, antihemorrhagic monitoring and antibiotic therapy.

Estimated total cost is approximately EUR 2000 per patient.

The device, equipped with the necessary fiducials, can instead be cold-sterilised with soluble germicidal powder and will therefore be reusable for all patients clinically eligible for stereotactic treatment of the cervix. Also, it is usable with a disposable barrier protection system which is represented by the medical condom.

This would lead to a reduction in the total cost of the entire procedure, in addition to the cost of the fiducials.

### 4.3. Quality Assurance

The quality assurance of 3D-printed devices used in radiotherapy has been the focus of several recent publications [19,20,21,22]. Even so, the recommended tests of geometry and density have not been performed on the 3D-printed components of the endovaginal applicator. To ensure the safe clinical implementation of the endovaginal applicator, it is essential to utilise biocompatible materials that comply with international standards [23]. Nevertheless, the biocompatibility of the finished product is contingent on numerous factors, including the biocompatibility of the material itself, the biocompatibility of the printing process, and the capacity for adequate cleaning to meet infection control requirements [24]. During the printing process, it has been demonstrated that brass nozzles can release lead into the 3D prints. Consequently, the utilisation of a dedicated stainless-steel nozzle is strongly recommended for clinical applications. Furthermore, the low heat distortion temperatures of PLA render them unsuitable for hot sterilisation [25]. A satisfactory method for decontamination may be found in the sterilisation of the printed components in a germicide cleaning solution, given that the 3D-printed endovaginal applicator components developed in this study were demonstrated to be unharmed by washing and soaking. It is imperative to verify the accurate positioning of the endovaginal applicator through the utilisation of pre-treatment imaging methodologies, such as x-ray imaging, given the well-documented adverse impact of positioning errors on target coverage and the heightened risk of toxicities from radiotherapy treatments [26,27].

A limit of this study is represented by the absence of dose comparison with other techniques (ICBT for suitable patients with or without MR; SBRT with CT-based conventional LINACs or MR-based LINACs with or without surface-guided radiation therapy-SGRT), examples of treatment plans and/or other clinical data and outcomes. These aspects are out of the scope of this paper and will be discussed in the proper study (STARBACS).

## 5. Conclusions

In view of the findings, it can be concluded that Sifilotto^®^ has been successful in achieving tumour tracking. This endovaginal device guarantees reproducibility and optimisation parameters for the setup of patients with LACC who are suitable for stereotactic treatments. It also enables a reduction in costs associated with a reduced consumption of biomedical materials and a reduction in total treatment time compared to intramucosal implant procedures of the fiducials. Further results will soon be presented regarding clinical outcomes in a wider range of patients.

## 6. Patents

This applicator has obtained a utility model patent in on the date of 7 March 2025, N. 202023000003144, released by “Ministero delle Imprese e del Made in Italy”.

## Figures and Tables

**Figure 1 curroncol-32-00354-f001:**
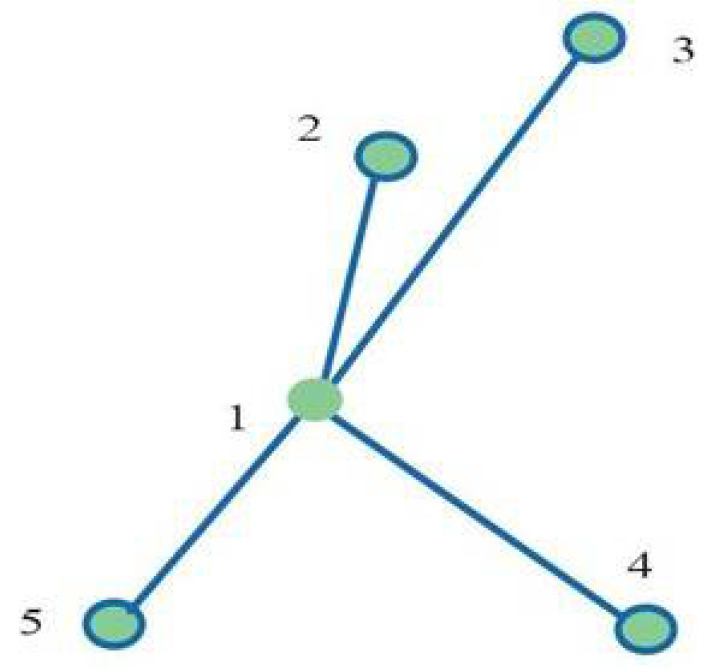
(**1**) Target centre; (**2**,**3**,**4**,**5**) spatial disposition of gold fiducials.

**Figure 2 curroncol-32-00354-f002:**
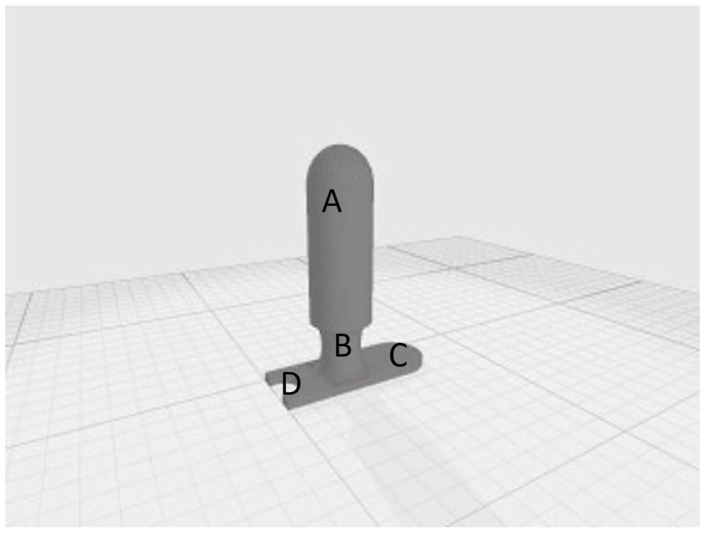
The 35 mm (**A**) cylinder, to be inserted into the vaginal canal, (**B**) neck, surrounded by the vulvar labia majora when the applicator in inserted, (**C**) applicator stopper, and (**D**) housing for the urinary catheter.

**Figure 3 curroncol-32-00354-f003:**
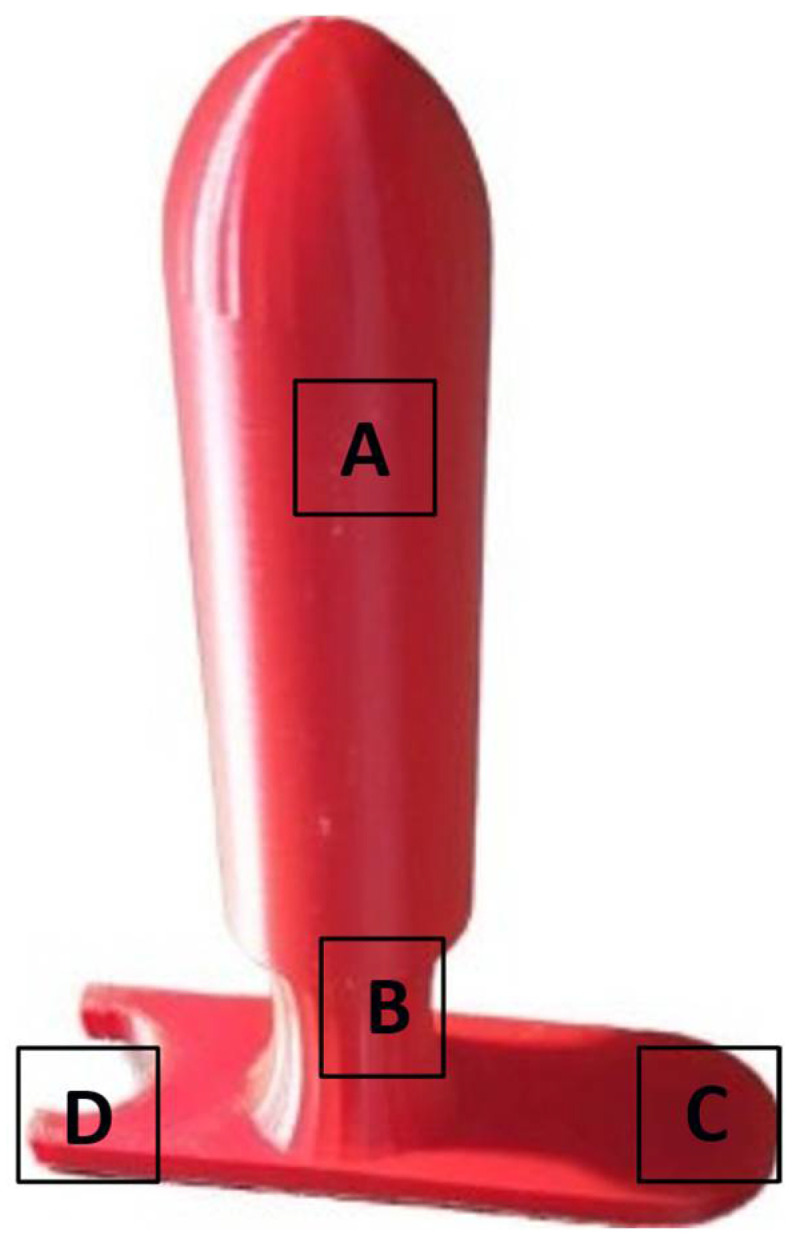
A picture of the 30 mm model Sifilotto^®^. (**A**) The cylinder, to be inserted in the vaginal canal; the (**B**) neck, surrounded by the vulvar labia majora when the applicator in inserted; (**C**) the applicator stopper; and the (**D**) housing for the urinary catheter.

**Figure 4 curroncol-32-00354-f004:**
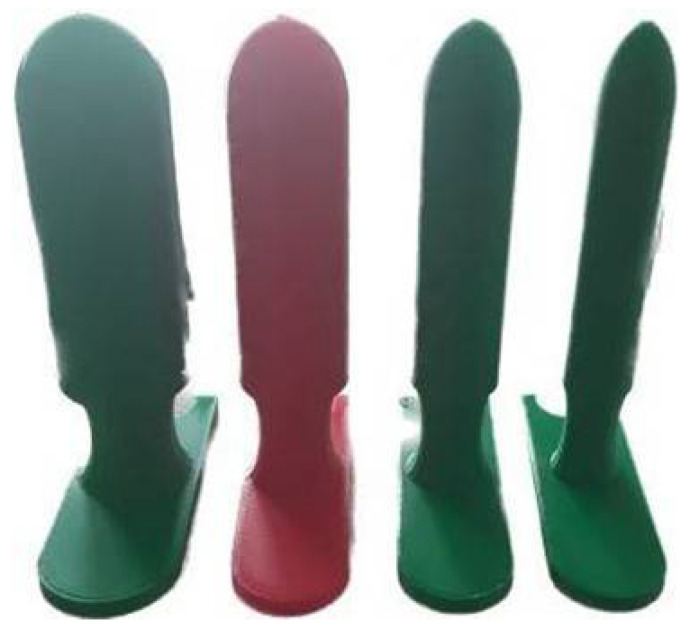
A picture of all the applicators: from left to right, 35 mm, 30 mm, 25 mm, 20 mm, 3D-printed (see Section 2).

**Figure 5 curroncol-32-00354-f005:**
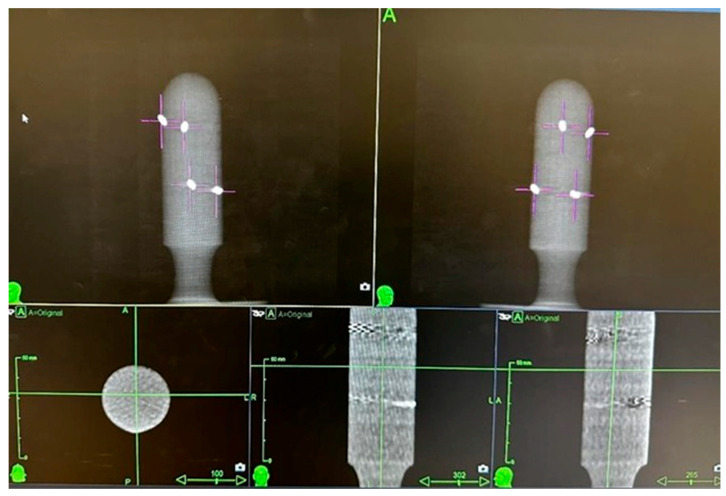
Referring to “Step 1”, 45° right and left view of the DRR extracted to the Treatment Planning System (**up**) and CT scans in axial, sagittal and coronal projection (**down**).

**Figure 6 curroncol-32-00354-f006:**
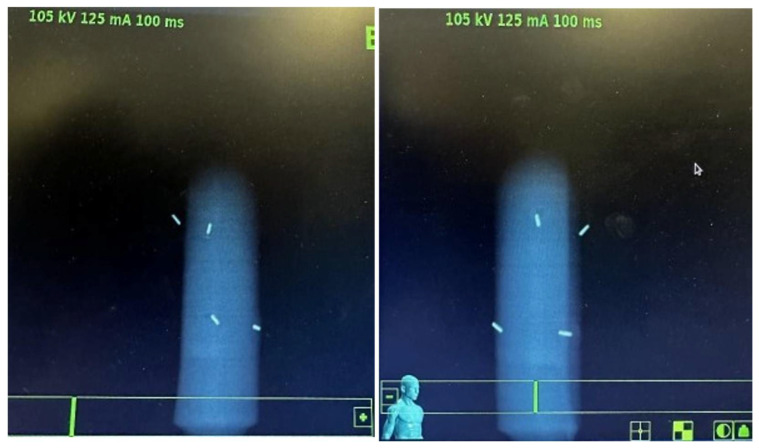
Referring to Step 2, x-ray images acquired with the dedicated oblique beam cameras.

**Figure 7 curroncol-32-00354-f007:**
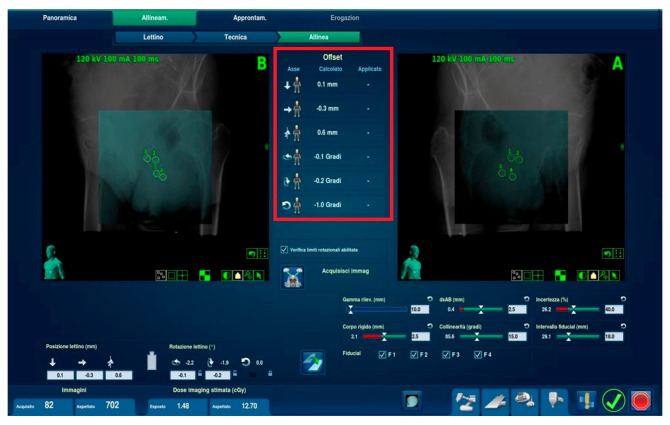
Fiducial tracking and suggested adjustment arrows in human body icons (red box in the centre) and x-rays/DRR matching in two oblique projections (left and right) with fiducial identification (green and yellow circles).

**Table 1 curroncol-32-00354-t001:** Subjective parameters collected for the seven patients. Range varies from 0 = not perceived to 10 = totally perceived.

Patient	Pain ^1^	Burning ^1^	Discomfort ^1^	Psychological Impact ^1^
1	5	7	4	4
2	3	2	3	3
3	2	2	3	2
4	4	4	2	2
5	3	3	1	2
6	2	2	1	1
7	2	7	2	2

^1^ Parameters are collected referring to points.

## Data Availability

No new data was created during this paper’s production. All available information about registered products are findable on official websites.

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
