# Peer review of "Design and Development of a Device (Sifilotto®) for Tumour Tracking in Cervical Cancer Patients Undergoing Robotic Arm LINAC Stereotactic Body Radiation Therapy Boost: Background to the STARBACS Study"

_curroncol, 2025, doi:10.3390/curroncol32060354_

Round 1

Reviewer 1 Report

Comments and Suggestions for Authors

Reviewer 2 Report

Comments and Suggestions for Authors

General Remarks:  This manuscript describes the development of a novel endovaginal applicator to localizing cervical cancer patient anatomy to receive a robotic-based stereotactic body radiation therapy boost in patients unable to tolerate brachytherapy boost therapy.   This is a preliminary study with only seven in vivo patients included in their study thus far.  The paper is mostly clearly written although some editorial review and proofreading by a native English language speaker is recommended.  I have some concerns of how a vaginal device is used since the target tissue for a cervical cancer is generally 2 cm above the cervical os.  The applicator itself is a modified cylindrical shape with internal fiducials so there are questions of rotational shifts between day to day treatments and daily imaging QA requirements to ensure the applicator is implanted correctly. 

My specific comments to be addressed by the authors are provided below.

Line 77:  Word “offer” should be replaced with “outcome”

Line 85:  Many institutions are now using MRI as well as CT scans for treatment planning.  The markers used in the device should be visible for localization and treatment planning and compatible with the application of either imaging modalities.

Paragraph 2.2:  How is the distance spacing and rotational orientation between fiducials accurately achieved when creating the applicator?  Are these implanted within the applicator itself?  What is the definition of “target centre?”  I am accustomed to this being a point relative to the patient anatomy, not some point within the applicator.  By the applicator design description, this appears to be a vaginal applicator, but the target for a cervical cancer is above the cervical os which is above the vagina.  I am confused at how this applicator is to be used in the patient by this description.  Why are the fiducials in the applicator and not implanted on the patient?  How do you distinguish localization error due to incorrect applicator translational or rotational positioning within the patient versus anatomical changes or patient positioning being the source of error prior to beam-on? 

Paragraph 3.1:  Radiological tests should include MRI as well since many institutions use MRI as the primary imaging modality for cervical cancers because of the immense improvement in soft tissue contrast for visually imaging the tumor volume.

Sentence starting Line 64 is a bit awkward as written

Line 95:  What are the sterilization procedures for the proposed device?  Most sterilization methods apply the use of high temperature heat which could cause deformations in the device?  I latter saw that this point is actually covered in the discussion.  Consider addressing this issue earlier in the paper.

Section 3.2:   The authors state that the SBRT doses were beyond the scope of this work.  However, it would be interesting to know if the typical doses used in the EMBRACE protocol was used as a starting point for the study.  If not, what did the authors use and why?

Line 246:  What are the average and maximal positional errors associated with using the Sifilotto® applicator compared to using established, conventional SBRT localization techniques?  Has this been assessed?

Comments on the Quality of English Language

Some awkward sentence and phrases detected in the text.

Reviewer 3 Report

Comments and Suggestions for Authors

Comments to curroncol-3612914

Entitled “Design and development of a device (Sifilotto®) for tumor tracking in cervical cancer patients undergoing robotic arm linac stereotactic body radiation therapy boost.”

General Comments:

  1. This paper describes the design and preliminary evaluation of a 3D-printed intravaginal applicator (Sifilotto®) to aid tumor tracking during SBRT boost in patients with locally advanced cervical cancer who are not candidates for intracavitary brachytherapy.
  2. The efforts to reduce the invasiveness of fiducial implantation are laudable, and the authors provide a technically detailed description of device development and feasibility testing. However, this approach's clinical value and implications remain to be fully addressed.

Specific Comments:

  1. Lack of evidence on dose assessment and target coverage:

The manuscript does not include any dose comparisons or data

to demonstrate whether the use of this device in SBRT is adequate to cover the residual cervical tumor volume.  Suggestions: The authors should provide examples of treatment planning, even if preliminary data, showing how SBRT boost using this device can achieve dose conformality and organ at risk (OAR) sparing. Without this, the clinical relevance of the device remains speculative.

  1. Limited rationale for invasive vaginal insertion in the SBRT era:

Modern SBRT platforms have advanced image guidance systems (e.g., CBCT, MRI-guided LINAC) that allow tumor localization without physically implanted tracking devices. It is unclear why the authors employed an intravaginal device that could increase patient discomfort, primarily when noninvasive alternatives exist.  Suggestions: The authors should discuss and justify why an internal tracking device is necessary in such a case. Considering current advances in image-guided radiation therapy, they should also compare their approach with other noninvasive tracking strategies.

  1. Insufficient clinical outcome data:

Despite using a small patient population (n = 7) for feasibility testing, the manuscript lacks follow-up data on treatment toxicity beyond tumor control, local recurrence, or patient-reported discomfort.  Suggestions: Even acknowledging the preliminary nature of the study, it would be helpful to present early clinical results or at least provide a follow-up timeline to assess tumor response and toxicity.

  1. This paper explores relevant technical challenges in gynecologic radiation oncology. However, the current research is mainly technical and proof-of-concept. Without dose validation and clinical outcomes data, it is too early to assess the impact of this device on patient care.

  1. Adding comparative doses, arguments against modern image-guided alternatives, and discussing the impact on clinical workflow would greatly strengthen the manuscript.

  1. In Figures 2,3 and 4, please add the functions of A, B, C, and D.

In Figure 4: How do you divide it into different sizes?

  1. How can these adapters be used if severe pelvic adhesion is caused by previous surgery or radiotherapy?

  1. In Table 1, the range varies are too large, if only from the patient’s subjective feeling. Why the burning parameter’s mean point is most high when compared with other parameters?

  1. The whole article needs to be polished and revised in English grammar

Comments on the Quality of English Language

The whole article needs to be polished and revised in English grammar

Round 2

Reviewer 3 Report

Comments and Suggestions for Authors

Comments to curroncol-3612914-peer-review-V3:

Specific comments:

  1. Scope Misalignment and Premature Publication:

The manuscript reads more like a patent-related technical report than a scientific article with clinical or translational relevance. While innovation is welcomed, publication in a peer-reviewed oncology journal typically requires evidence of clinical impact, which is lacking in this case. The authors note that results will be presented in the future STARBACS study but that deferral limits the value of the current submission.

  1. Potential Conflict of Interest:

As the device is patented and commercial interests may be involved, a clearer discussion of conflicts of interest and steps taken to ensure unbiased reporting is warranted.

  1. Conclusion: 

The manuscript does not provide convincing evidence that the device fulfills a substantial clinical need or offers significant improvements over current treatment modalities.

Author Response

Comment 1:

  1. Scope Misalignment and Premature Publication:

The manuscript reads more like a patent-related technical report than a scientific article with clinical or translational relevance. While innovation is welcomed, publication in a peer-reviewed oncology journal typically requires evidence of clinical impact, which is lacking in this case. The authors note that results will be presented in the future STARBACS study but that deferral limits the value of the current submission.

Response 1:

The manuscript is a descriptive study reporting the fundamental preliminary phase of the STARBACS study. We regret that the reviewer found the paper unappealing; however, we would like to highlight a previous experience involving a patented device:

Kubota Y, Ohno T, Kawashima M, Murata K, Okonogi N, Noda SE, Tsuda K, Sakai M, Tashiro M, Nakano T. Development of a Vaginal Immobilization Device: A Treatment-planning Study of Carbon-ion Radiotherapy and Intensity-modulated Radiation Therapy for Uterine Cervical Cancer. Anticancer Res. 2019 Apr;39(4):1915-1921. doi: 10.21873/anticanres.13300. PMID: 30952733.

That study includes dosimetric data; however, in all cases, patients could undergo both IMRT and CIRT. In contrast, our patients have no guideline-supported alternatives if they do not accept IBRT. Therefore, we believe the most meaningful contribution we can offer to the scientific community lies in the clinical outcomes.

To obtain robust clinical results, an adequate follow-up period and sufficient sample size are necessary. We have clearly stated that clinical outcomes fall outside the scope of this preliminary study, and we acknowledge the absence of clinical data as a limitation. Within the framework of peer review, such an approach is acceptable

Comment 2:

  1. Potential Conflict of Interest:

As the device is patented and commercial interests may be involved, a clearer discussion of conflicts of interest and steps taken to ensure unbiased reporting is warranted. 

Response 2:

Conflict of interest are declared in the manuscript and explicited in the requested Susy MDPI form. As a specific response to the reviewer: the entire research group is totally auto-financed, nor company has contributed to the scientific project. 

Comment 3:

  1. Conclusion: 

The manuscript does not provide convincing evidence that the device fulfills a substantial clinical need or offers significant improvements over current treatment modalities

Response 3:

The reviewer may consider this device to be of limited relevance in daily clinical practice; however, other researchers worldwide might view it as a valuable milestone in this field. As we have stated, this is a fundamental preliminary phase of the broader STARBACS study, which will focus on clinical outcomes. We believe that this initial component can enhance the study’s reproducibility and clarity

Round 3

Reviewer 3 Report

Comments and Suggestions for Authors

The authors addressed all the questions raised by the reviewer.

Comments on the Quality of English Language

The English could be improved after the revised